# Influence of *BoLA-DRB3* Polymorphism and Bovine Leukemia Virus (BLV) Infection on Dairy Cattle Productivity

**DOI:** 10.3390/vetsci10040250

**Published:** 2023-03-27

**Authors:** Ayumi Nakatsuchi, Yasunobu Matsumoto, Yoko Aida

**Affiliations:** 1Research and Development Section, Institute of Animal Health, JA Zen-Noh (National Federation of Agricultural Cooperative Associations), 7 Ohja-machi Sakura-shi, Chiba 285-0043, Japan; 2Laboratory of Global Infectious Diseases Control Science, Graduate School of Agricultural and Life Sciences, The University of Tokyo, 1-1-1 Yayoi, Bunkyo-ku, Tokyo 113-8657, Japan; 3Laboratory of Global Animal Resource Science, Graduate School of Agricultural and Life Sciences, The University of Tokyo, 1-1-1 Yayoi, Bunkyo-ku, Tokyo 113-8657, Japan

**Keywords:** BLV infection, *BoLA-DRB3*, milk, productivity, milk trait, susceptible, resistant

## Abstract

**Simple Summary:**

Genetic selection and breeding of bovine leukemia virus (BLV)- susceptible and -resistant cattle based on polymorphisms within the bovine major histocompatibility complex (MHC), namely, bovine leukocyte antigen (*BoLA*)*-DRB3*, is important to control horizontal and vertical transmission of BLV. However, its effect on dairy cattle productivity is unknown. Here, we evaluated the effects of BLV infection and *BoLA-DRB3* on production performances such as milk yield. BLV infection significantly affected milk yield; however, *BoLA-DRB3* had no effect on dairy cattle productivity, suggesting that BLV infection affects dairy productivity more than genetic selective breeding. Our results indicate that genetic selective breeding of resistant cattle, or the preferential culling of susceptible dams, is a promising approach to developing an effective BLV eradication program.

**Abstract:**

Enzootic bovine leukosis caused by the bovine leukemia virus (BLV) results in substantial damage to the livestock industry; however, we lack an effective cure or vaccine. *BoLA-DRB3* polymorphism in BLV-infected cattle is associated with the proviral load (PVL), infectivity in the blood, development of lymphoma, and in utero infection of calves. Additionally, it is related to the PVL, infectivity, and anti-BLV antibody levels in milk. However, the effects of the *BoLA-DRB3* allele and BLV infection on dairy cattle productivity remain poorly understood. Therefore, we investigated the effect of BLV infection and *BoLA-DRB3* allele polymorphism on dairy cattle productivity in 147 Holstein dams raised on Japanese dairy farms. Our findings suggested that BLV infection significantly increased milk yield. Furthermore, the *BoLA-DRB3* allele alone, and the combined effect of BLV infection and the *BoLA-DRB3* allele had no effect. These results indicate that on-farm breeding and selection of resistant cattle, or the preferential elimination of susceptible cattle, does not affect dairy cattle productivity. Additionally, BLV infection is more likely to affect dairy cattle productivity than *BoLA-DRB3* polymorphism.

## 1. Introduction

The bovine leukemia virus (BLV) belongs to the *Deltaretrovirus* genus of the *Retroviridae* family. BLV is closely related to the human T-cell leukemia virus and is the causative agent of enzootic bovine leukosis (EBL), the most common neoplastic disease in cattle [1]. BLV is prevalent in most regions worldwide except for certain countries, such as Finland, Ireland, Spain, and Denmark [2]. In Japan, 40.9% of dairy cattle are infected with BLV [3].

In BLV-infected cattle, approximately 70% are asymptomatic and 30% are diagnosed with persistent lymphocytosis, some of which develops into EBL [1]. In addition to causing EBL, BLV infection is associated with lower milk and meat production and a shorter production lifespan [4,5,6,7]. Additionally, BLV infection can directly impair the immune system, predisposing animals to other opportunistic infections and diseases [8]. Thus, BLV infection causes economic damage owing to direct cattle culling associated with EBL, as well as silent economic damage unnoticeable to farmers. Despite its prevalence, we lack practical and effective treatments and vaccines against BLV. Additionally, the influence of BLV infection on the various milk traits of Holstein Friesian cattle is yet to be investigated.

The highly polymorphic major histocompatibility complex (MHC) is important for antigen presentation and immune response [9]. In cattle, the MHC system is known as the bovine leukocyte antigen (BoLA) complex. *BoLA-DRB3* locus within the BoLA class II subregion is associated with diseases in cattle. It is a well-studied locus that is highly polymorphic and functionally important. To the best of our knowledge, 384 DRB3 alleles have been listed in the MHC database of the IPD-MHC (https://www.ebi.ac.uk/ipd/mhc/group/BoLA/ (accessed on 1 January 2023). Polymorphisms in this region are associated with individual differences in an immune response to a particular infectious disease, such as mastitis [10], tick-borne disease [11,12], foot and mouth disease [13], bovine herpesvirus 1 [14], and bovine papillomavirus-induced bladder cancer [15]. Moreover, *Bo-LA-DRB3* polymorphism affects dairy cattle productivity such as milk quality and production rate during a mastitis infection [16], the microbiota in colostrum and milk [17], as well as reproduction rates in neosporosis [18]. Particularly, *BoLA-DRB3* is reportedly associated with mastitis and somatic cell count (SCC) [19,20,21,22,23,24].

*BoLA-DRB3* polymorphism in BLV-infected cattle is associated with the proviral load (PVL), infectivity in the blood, development of lymphoma, and in utero infection of calves [25,26,27,28,29,30,31,32]. For example, cattle with the *BoLA-DRB3*015:01* and *DRB3*012:01* alleles are more infectious and have higher PVL, which increases the risk of horizontal transmission [29]. In contrast, cattle with the *BoLA-DRB3*009:02*, *DRB3*014:01:01* [25,31,32], and *DRB3*002:01* alleles [25] were at a lower risk of horizontal transmission of BLV due to their low BLV infectivity and a low PVL [28]. Additionally, neutral cattle with other *BoLA-DRB3* alleles had no significant association with PVL in vivo [26]. Notably, we reported that susceptible, neutral, and resistant BLV-infected dams have varying levels of PVL, infectivity, and anti-BLV antibody in milk [33,34]. However, we lack comprehensive information on whether *BoLA-DRB3* alleles may affect dairy cattle productivity such as milk yield, milk production, and compositional quality traits in BLV-infected Holstein cattle. Therefore, we aimed to evaluate whether the presence of BLV infection and differences in the *BoLA-DRB3* locus can affect dairy cattle productivity in Holstein cattle from Japan.

## 2. Materials and Methods

### 2.1. Blood Sample Collection and Genomic DNA Isolation

This study was approved by the Animal Ethics Committee and the Animal Care and Use: Animal Experiments Committee of the Graduate School of Agricultural and Life Sciences, at the University of Tokyo (Approval Number p22–2–030).

Blood samples were collected from 147 Holstein dams from three different dairies in the Chiba prefectures in Japan and stored in tubes containing ethylenediaminetetraacetic acid (EDTA) and tubes for serum separation (Table 1). The serum was separated from the blood samples to detect BLV antibodies. DNA was extracted using the Wizard^®^ Genomic DNA Purification Kit (Promega Corporation, Madison, WI, USA) from the blood samples according to the manufacturer’s instructions.

### 2.2. Diagnosis of BLV Infection

BLV infection in cattle was confirmed by detecting anti-BLV antibodies in serum using a commercial ELISA kit (Nippon Gene, Tokyo, Japan) and by detecting the BLV provirus in DNA using the CoCoMo™-BLV Primer/Probe (Nippon Gene, Tokyo, Japan) with THUNDERBIRD Probe qPCR Mix (Toyobo, Tokyo, Japan). Samples that tested positive for either the anti-BLV antibody or the provirus were considered “BLV-infected”.

### 2.3. BoLA-DRB3 Allele Typing

*BoLA-DRB3* was identified by polymerase chain reaction (PCR)-sequence-based typing (SBT) method using genomic DNA from blood [25,26]. Briefly, PCR was performed using the DRB3 forward (5′-CGCTCCTGTGAYCAGATCTATCC-3′) and DRB3 reverse (5′-CACCCCCGCGCTCACC-3′) primer set to amplify *BoLA-DRB3* exon 2. PCR fragments were purified and sequenced using the BigDye™ Terminator v1.1 Cycle Sequencing Kit (Thermo Fisher Scientific, Waltham, MA, USA). The sequencing data were then analyzed to determine the *BoLA-DRB3* allele using ASSIGN 400 AFT software (Conexio Genomics, Fremantle, Australia).

### 2.4. Assessment of Dairy Cattle Productivity

The milk information including milk yield, fat percentage, protein percentage, percent of non-fat solids, SCC, and milk urea nitrogen (MUN) was analyzed by the Livestock Improvement Association of Japan and provided by each farmer.

### 2.5. Staristical Analysis

Significant differences among multiple groups were identified using the least-squares variance analysis method and Tukey’s multiple comparison test. Statistical significance was set at *p*  <  0.05. All tests were performed using the R statistical package version 4.1.2.

## 3. Results

### 3.1. Determination of the BoLA-DRB3 Alleles and BLV Infection

To evaluate the effects of BLV infection and *BoLA-DRB3* polymorphism on the dairy cattle productivity, we investigated the BLV infection status and *BoLA-DRB3* alleles of 147 Holstein dams from three dairy farms in Japan (Table 1). BLV prevalence in each dairy farm ranged from 45.5–97.6% (mean 62.6%). The PCR-SBT method was used to typify the *BoLA-DRB3* allele. PCR-SBT identified 14 known *BoLA-DRB3* alleles (*BoLA-DRB3*001:01*, **002:01*, **006:01*, **007:01*, **007:04*, **009:02*, **010:01*, **011:01*, **012:01*, **014:01:01*, **015:01*, **016:01*, **018:01*, and **027:03*) in the IPD-MHC database. Using *BoLA-DRB3* allele data, 147 dams were divided into three groups: susceptible, resistant, and neutral dams (Table 1). Susceptible dams identified as carrying at least one susceptible *BoLA-DRB3*012:01* or *DRB3*015:01* allele and had high BLV PVL [29], resistant dams identified as carrying at least one resistant *BoLA-DRB3*002:01* [25], *DRB3*009:02* [25,31,32], or *DRB3*014:01:01* [25,31,32] alleles and had low BLV PVL, and neutral cattle carrying other *BoLA-DRB3* alleles in their genome. Because of the predominance of the resistance trait over the susceptibility trait, dams carrying both resistance and susceptibility alleles were defined as BLV resistant [26]. BLV PVL in the blood of BLV-positive cattle was compared among susceptible, neutral, and resistant dams from the samples used in this study. The mean PVL values in blood were 32,247, 15,006, and 11,798 copies/10^5^ cells for the susceptible, neutral, and resistant groups, respectively (Figure 1). These results indicated that the blood PVL of susceptible dams was significantly higher than that of the neutral (*p* = 0.0064) and resistant dams (*p* = 0.0014), which is consistent with previous reports [28,33] (Figure 1).

### 3.2. Distribution of Dams Based on Calving Number and Lactation Period

Differences in farms, calving numbers, and lactation periods might affect dairy cattle productivity [35,36]. Accordingly, we evaluated the distribution of dams by calving number and lactation period (Figure 2). The 147 dams had calving numbers ranging from 1–7, with an average calving number of 2.0 (Figure 2A). Based on the report of Strucken et al., we further divided the dam into four lactation stages (Figure 2B): phase 1, 1–50 days (early lactation); phase 2, 51–110 days (peak lactation); phase 3, 111–210 days (mid-lactation); and phase 4, 211 days to dry period (late lactation) [37]. Approximately two-thirds of the dams were in phase 3 or 4, with an average lactation period of 3.0.

### 3.3. Effect of BLV Infection on Dairy Cattle Productivity

Although the effect of BLV infection on milk yield was evaluated previously [7,38], the effect of this infection on other milk-related dairy cattle productivity has rarely been evaluated. Therefore, in addition to milk yield, the effect of a BLV infection on milk fat percentage, protein percentage, non-fat solids percentage, SCC, and MUN was evaluated (Figure 3). In conducting this analysis, we considered the possibility that farm differences, calving numbers, and lactation periods could affect dairy cattle productivity and used a linear model to evaluate these effects. Factors that were found to be significantly different were then added to the analytical model and examined using the least squares analysis of variance method [39]. Among the dairy cattle productivity examined in this study, BLV-positive dams had significantly higher yields than BLV-negative dams (*p* = 0.049). Fat percentage, protein percentage, non-fat solids percentage, SCC, and MUN were not significantly different.

### 3.4. BoLA-DRB3 Polymorphism Does Not Affect Dairy Cattle Productivity

There is little information regarding whether *BoLA-DRB3* polymorphism on its own may affect dairy cattle productivity such as milk yield, milk production, and compositional quality traits in Holstein cattle. Therefore, we evaluated possible differences in dairy cattle productivity among susceptible, neutral, and resistant dams (Table 1). Among the genotypes examined in this study, no significant differences were observed in the percentages of fat, protein, or non-fat solids, nor in milk yield, SCC, or MUN (Table 2).

### 3.5. Combined Effect of BoLA-DRB Polymorphism and BLV Infection on Dairy Cattle Productivity

Our findings indicated that BLV infection affects dairy cattle productivity, whereas there were no differences in these percentages among susceptible, neutral, and resistant dams. As we lack results on the combined effect of *BoLA-DRB3* polymorphism and BLV infection on milk traits, we evaluated the effects of BLV infection and *BoLA-DRB3* polymorphism on the dairy cattle productivity. Among the parameters examined in this study, no significant differences were observed for all of them (Figure 4).

## 4. Discussion

In this study, we investigated the effects of BLV infection and *BoLA-DRB3* polymorphism on dairy cattle productivity. We observed the following: First, BLV infection may increase milk yield. Second, based on *BoLA-DRB3* polymorphism, no significant differences were observed in dairy cattle productivity among BLV-susceptible, -neutral, and -resistant dams. These results contrast with previous reports that evaluated the effect of *BoLA-DRB3* on dairy cattle productivity when accompanied by diseases such as mastitis [19,21]. This indicates that *BoLA-DRB3* may not affect dairy cattle productivity in the absence of disease. Third, analyzing the combined effect of BLV infection and *BoLA-DRB3* on dairy cattle productivity revealed no significant differences in all parameters. To the best of our knowledge, this is the first study reporting the combined effect of both BLV infection and *BoLA-DRB3* on dairy cattle productivity. *BoLA-DRB3* polymorphisms are reportedly considered valuable in controlling the risk of horizontal and vertical transmission of BLV [16,27,28]. In addition, our results indicate that new *BoLA-DRB3*-specific measures to increase the number of resistant dams and eliminate preferentially susceptible dams may not affect dairy cattle productivity.

In this study, BLV infection had a positive effect on milk yield when analyzed considering farm differences, calving number, and lactation period. An experimental study of BLV infection reported that BLV-positive dams produced more milk, which is consistent with our report [38]. Additionally, a field study reported that herds with higher BLV prevalence actually produce more milk [40]. On the other hand, some field studies have reported reduced milk production in BLV-infected dams [6,7,41,42], while others have reported no association between BLV infection and milk production [43]. However, even among the reports about decreased milk production induced by BLV infection, the number of calvings at which milk production decrease varied, with some dams having decreased milk production at two or more calvings [7], some having decreased milk production at three or more calvings [41], some having decreased milk production at four or more calvings [42], and others having decreased milk production at two or three calvings but with no difference in milk production at four or more calvings [6]. Several factors such as study size, housing environment, and geography may have influenced these differences. Experimental infection studies with susceptible and resistant dams may be needed to verify the accuracy of the results of these studies. 

No significant differences in dairy cattle productivity were observed between susceptible, neutral, and resistant dams after accounting for differences in farm, number of calves, and lactation period. While previous studies have evaluated the impact of each *BoLA-DRB3* allele on dairy cattle productivity, we evaluated the impact of genotypes that may effectively prevent the spread of BLV infection in the present study [19,20,21,22,23,24]. Differences in the analytical methods used likely may have affected the results. In addition, our sample size may not have been sufficient, and we believe that a larger sample size is needed for a more accurate assessment.

The effects of BLV infection and *BoLA-DRB3* polymorphism on dairy cattle productivity were previously independently evaluated. However, when considering the use of *BoLA-DRB3* as one of the indicators to prevent the spread of BLV infection, it is essential to evaluate the impact of both BLV infection and *BoLA-DRB3* on dairy cattle productivity. Therefore, we evaluated the dairy cattle productivity under both BLV infection and *BoLA-DRB3*, taking into account differences in farm, calving number, and lactation period, and found no significant differences in dairy cattle productivity traits. To date, there have been no reports that susceptible dams are more likely to be infected with BLV than resistant dams. On the other hand, we have reported previously that some resistant dams are less likely to develop EBL, while susceptible dams are more likely to develop high PVL [29] and EBL [44]. Unfortunately, this study targeted the productivity of BLV-infected dams without lymphoma, but did not investigate the productivity of BLV-infected dams with lymphoma. Therefore, further study is required to define the relationship between economic traits and polymorphism of *BoLA-DRB3* in BLV-infected cattle with advanced symptoms. To our knowledge, this is the first report examining the combined effects of BLV infection and *BoLA-DRB3*.

Several studies have reported the impact of BLV infection and *BoLA-DRB3* on cattle productivity [4,5,6,7,19,20,21,22,23,24,38,40,41,42,43]. Results from previous studies differed with regards to BLV infection, where it was found to have positive, negative, as well as no effect on cattle productivity [4,5,6,7,38,40,41,42,43]. Additionally, *BoLA-DRB3* reportedly has positive or negative effects on dairy cattle productivity [19,20,21,22,23,24]. Our results indicated that BLV infection might increase milk yield; however, after accounting for the effect of *BoLA-DRB3*, no significant difference in milk yield was observed. These results indicate that BLV infection impacts dairy cattle productivity considerably more than *BoLA-DRB3*. Therefore, a breeding strategy for BLV control using *BoLA-DRB3*, such as culling susceptible dams and actively breeding resistant dams, would not have a negative impact on milk production. Resistant dams have a significantly lower risk of horizontal infection [28], in utero infection [27], and milk transmission [33] than susceptible dams, suggesting that increasing the proportion of resistant dams may reduce transmission risk. In contrast, the relationship between *BoLA-DRB3* and reproductive performance, which is as important as dairy cattle productivity, has not been well studied. Therefore, to confirm the detailed effect of *BoLA-DRB3* on reproductive performance, future studies regarding the genetic background of cows with various traits, such as high-yielding dairy cattle and cows with low productivity are indispensable.

## 5. Conclusions

In this study, we investigated the effect of BLV infection and *BoLA-DRB3* polymorphism on dairy cattle productivity. Our findings confirmed that dairy cattle productivity is affected by BLV infection rather than *BoLA-DRB3* polymorphism. Therefore, our results would aid in developing effective BLV eradication programs, such as the selective breeding of cattle with BLV-resistant *BoLA-DRB3* alleles and the preferential elimination of cattle with susceptible *BoLA-DRB3* alleles.

## Figures and Tables

**Figure 1 vetsci-10-00250-f001:**
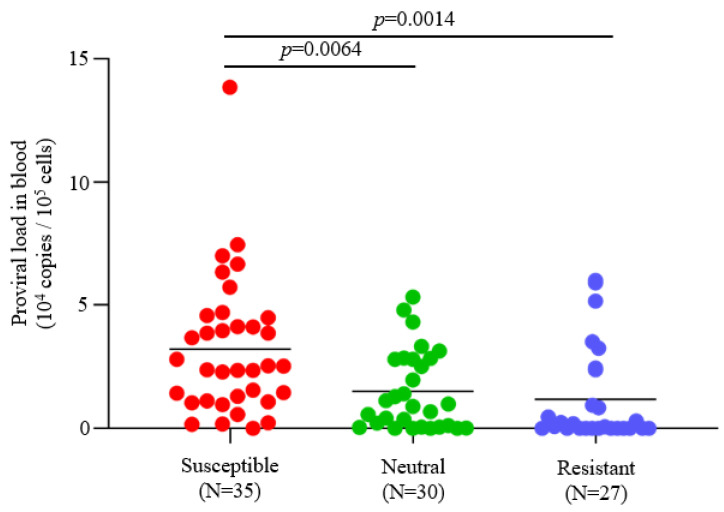
Comparison of proviral load (PVL) in blood from BLV-positive dams. BLV infection was determined by a combination of the BLV Env gp51 specific antibodies detection and provirus detection by BLV-CoCoMo-qPCR-2. Blood samples were obtained from 92 BLV-positive dams and extracted DNAs. The PVLs in blood were measured using the CoCoMo-qPCR-2 method (Nippon Gene, Tokyo, Japan). *BoLA-DRB3* alleles were typed by the PCR-SBT method using DNA from blood. All BLV-positive dams were divided into resistant, susceptible, and neutral groups based on the presence of *BoLA-DRB3* alleles as follows: susceptible dams carried at least one *BoLA-DRB3*012:01* or **015:01* allele in their genomes; resistant dams carried at least one *BoLA-DRB3*002:01*, **009:02*, or **014:01:01* allele in their genomes; and neutral dams carried other alleles in their genomes. Dams carrying both susceptible and resistant alleles were defined as resistant. N = number of tested dams. Mean PVL values were compared among groups using Tukey’s multiple comparison test.

**Figure 2 vetsci-10-00250-f002:**
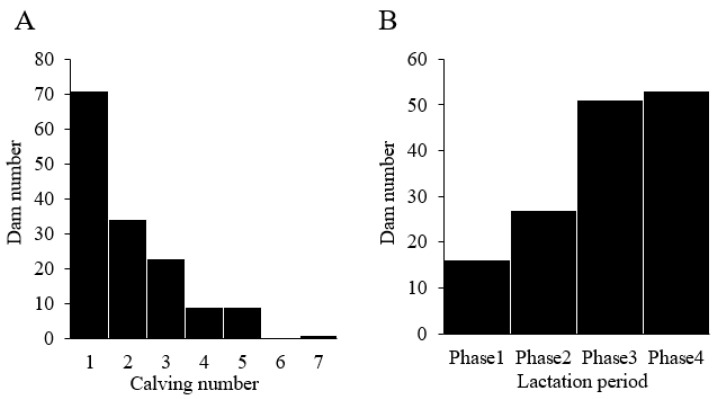
Distribution of dams by calving number and lactation period. (**A**) Distribution of dams by calving number. (**B**) Distribution of dams by lactation period. Dams were divided into four lactation phases: phase 1, 1–50 days (early lactation); phase 2, 51–110 days (peak lactation); phase 3, 111–210 days (mid-lactation); and phase 4, 211 days—dry period (late lactation).

**Figure 3 vetsci-10-00250-f003:**
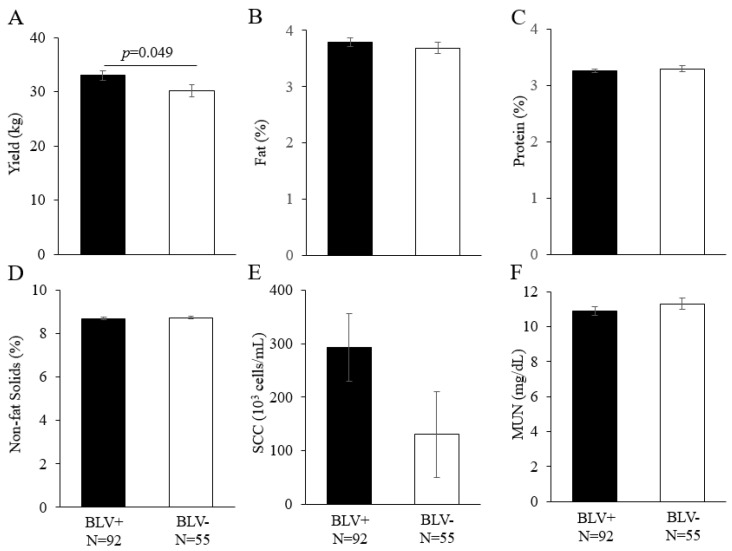
Effects of BLV infection on dairy cattle productivity ((**A**) yield (kg), (**B**) fat (%), (**C**) protein (%), (**D**) non-fat solids (%), (**E**) SCC (10^3^ cells/mL), and (**F**) MUN (mg/dL)). The results of the linear model analysis showed significant differences in the following factors: (**A**) lactation period; (**B**) farm and lactation period; (**C**) farm, lactation period, and calving number; (**D**) farm, lactation period, and calving number; (**E**) none; (**F**) farm and lactation period. A least-squares analysis of variance was performed in addition to the factors for which significant differences were found. BLV, bovine lymphoma virus; MUN, milk urea nitrogen; SCC, somatic cell count. BLV infection was determined by a combination of the BLV Env gp51 specific antibodies detection and provirus detection by BLV-CoCoMo-qPCR-2. BLV+ = BLV positive dam; BLV−, BLV negative dam; N, number of tested dams.

**Figure 4 vetsci-10-00250-f004:**
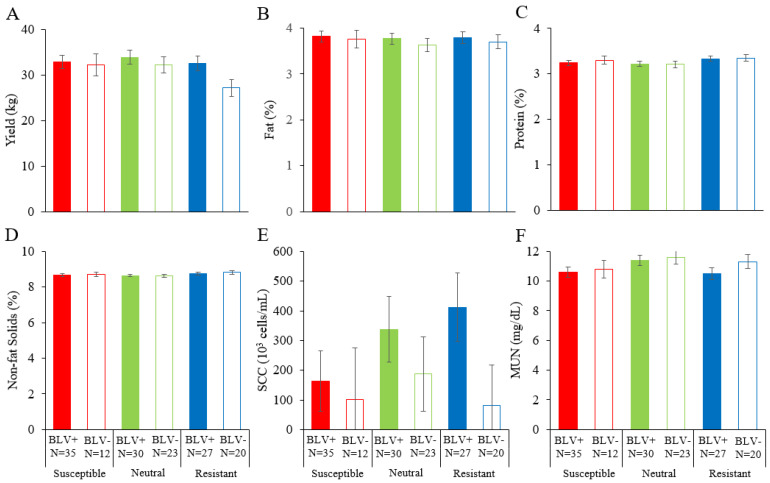
Effects of both BLV infection and *BoLA-DRB3* on dairy cattle productivity ((**A**) yield (kg), (**B**) fat (%), (**C**) protein (%), (**D**) non-fat solids (%), (**E**) SCC (10^3^ cells/mL), and (**F**) MUN (mg/dL)). The results of the linear model analysis showed significant differences in the following factors: (**A**) lactation period; (**B**) farm and lactation period; (**C**) farm, lactation period, and calving number; (**D**) lactation period and calving number; (**E**) none; (**F**) farm and lactation period. A least-squares analysis of variance was performed in addition to the factors for which significant differences were found. BLV, bovine lymphoma virus; MUN, milk urea nitrogen; SCC, somatic cell count. BLV infection was determined by a combination of the BLV Env gp51 specific antibodies detection and provirus detection by BLV-CoCoMo-qPCR-2. BLV+ (■, ■, ■), BLV positive dam; BLV− (□, □, □), BLV negative dam; N, number of tested dams.

**Table 1 vetsci-10-00250-t001:** Sample number, bovine leukemia virus (BLV) prevalence, and distribution of susceptible, neutral, and resistant dams.

Farm	Total Dam	Susceptible Dam ^b^		Neutral Dam ^c^		Resistant Dam ^d^	
		BLV+ ^a^ (%)	BLV− ^a^ (%)	BLV+ ^a^ (%)	BLV− ^a^ (%)	BLV+ ^a^ (%)	BLV− ^a^ (%)
A	88	15 (17.0)	10 (11.4)	15 (17.0)	20 (22.7)	10 (11.4)	18 (20.5)
B	42	19 (45.2)	0 (0.0)	7 (16.7)	1 (2.4)	15 (35.7)	0 (0.0)
C	17	1 (5.8)	2 (11.8)	8 (47.0)	2 (11.8)	2 (11.8)	2 (11.8)
Total	147	35 (23.8)	12 (8.2)	30 (20.4)	23 (15.6)	27 (18.4)	20 (13.6)

^a^ BLV infection was determined by a combination of the BLV Env gp51 specific antibodies detection and provirus detection by BLV-CoCoMo-qPCR-2. BLV+, BLV positive dam; BLV−, BLV negative dam. ^b,c,d^ PCR sequence-based typing was used to identify *BoLA-DRB3* alleles. ^b^ Dams carried at least one susceptible *BoLA-DRB3*012:01* or **015:01* allele, but no resistant allele. ^c^ Dams had neither susceptible nor resistant alleles. ^d^ Dams carried at least one resistant *BoLA-DRB3* 002:01*, **009:02*, or **014:01:01* allele.

**Table 2 vetsci-10-00250-t002:** Effects of *BoLA-DRB3* on dairy cattle productivity.

Milk Parameter	*BoLA-DRB3*	Estimated Coefficients	Standard Error for the Coefficient	*p*-Value
Yield (kg)	Susceptible vs. Neutral	0.271	1.66	0.9855
	Susceptible vs. Resistant	2.432	1.73	0.3395
	Neutral vs. Resistant	2.703	1.67	0.2432
Fat (%)	Susceptible vs. Neutral	0.0939	0.127	0.7421
	Susceptible vs. Resistant	0.0500	0.127	0.9183
	Neutral vs. Resistant	0.0439	0.126	0.9355
Protein (%)	Susceptible vs. Neutral	0.0421	0.0570	0.7421
	Susceptible vs. Resistant	0.0831	0.0579	0.3264
	Neutral vs. Resistant	0.1252	0.0567	0.0735
Non-fat Solids (%)	Susceptible vs. Neutral	0.0501	0.0706	0.7588
	Susceptible vs. Resistant	0.0919	0.0717	0.4079
	Neutral vs. Resistant	0.1420	0.0702	0.1108
SCC (10^3^ cells/mL)	Susceptible vs. Neutral	125.2	121	0.5583
	Susceptible vs. Resistant	124.3	125	0.5811
	Neutral vs. Resistant	0.837	121	1.000
MUN (mg/dL)	Susceptible vs. Neutral	0.827	0.391	0.0910
	Susceptible vs. Resistant	0.183	0.390	0.8863
	Neutral vs. Resistant	0.644	0.387	0.2234

The results of the linear model analysis showed significant differences in the following factors: (A) lactation period; (B) farm and lactation period; (C) farm, lactation period, and calving number; (D) farm, lactation period, and calving number; (E) none; (F) farm and lactation period. A least-squares analysis of variance was performed in addition to the factors for which significant differences were found.

## Data Availability

The data presented in this study are available on request from the corresponding author.

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
