# Peer review of "Influence of BoLA-DRB3 Polymorphism and Bovine Leukemia Virus (BLV) Infection on Dairy Cattle Productivity"

_vetsci, 2023, doi:10.3390/vetsci10040250_

Round 1

Reviewer 1 Report

The Authors have taken a new insight by considering both BLV infections and BoLA-DRB3 polymorphism in terms of dairy cattle productivity. In all, this is a well written article that has got  support to be published.  There are only  some minor comments:  

Line 90 : Is there any particular reason why farms from Chiba prefecture were chosen?

Line 132: What was the picture in terms of BLV PVL levels among neutral cattle carrying other BoLA-DRB3 alleles?

Table 1 – BLV prevalence we know generally on farm, and how it evolved in different groups of dams: Susceptible, neutral and resistant?

Reviewer 2 Report

Well written paper with a clear presentation of methods and results. However, already in the introduction, confusion arises around the effects of BLV infection on milk yield. The abstract says it increases (line 31), introduction (line 48) says decrease and in the discussion, it appears that reports go in two directions.

typo: Line 44 Fin-land

In the results, I miss the quantification of the PVL. It would add to the quality of the paper and the results if the susceptible genotype can be related to higher PVL especially because analyses were made over multiple parities and phases in lactation.

Line 200: no differences were observed between any of them? this line is more or less repeated in line 218.

the discussion is a spaghetti, that goes back and forth between previous studies on BLV and genotype selection, and effects on productivity. The conclusions from the data in this study are draw correctly (first paragraph), but afterwards I get lost. If such contrasting outcomes are reported, one may have to discuss why that may be (study size, setup) in a structured way. Now dry matter intake is mentioned, this is not a strong part of the discussion. Maybe a conclusion is that different studies lead to different conclusions and that in the end no real effect can be distinguished. 

What are the risks of selection versus the benefits, bearing in mind that the effects on milk yield may indeed be minimal?

Reviewer 3 Report

This manuscript attempts to correlate BLV disease status and BoLA-DRB3 haplotype with milk production metrics to determine the economic impact of BLV resistance. The authors improved this manuscript from their first submission considerably, however some gaps in understanding of the study still remain.

1.      The authors describe the use of both ELISA and PCR for determining BLV disease status, however all figures only describe two categories of animals BLV + or BLV -. Is this status determined by ELISA or PCR? It is likely ELISA, and if so, why only segregate animals by ELISA status. Providing a breakdown of animal performance by PCR status would likely be more illuminating as resistant animals should not possess high PVLs and be more informative to the value of genetic BLV resistance. Are all BLV+ resistant animals PCR positive as well? Please address.

2.      BLV+ animals regardless of genotype appear to produce more milk over an entire lactation. If ELISA is being used to determine BLV status, then wouldn’t susceptible animals, more likely to become infected and progress in disease be more productive than resistant animals? Please address.

Round 2

Reviewer 2 Report

Thanks you for addressing the points raised in the initial review. The manuscript quality improved significantly.

line 248: an experimental study suggested?

line 250: a field study demonstrated?

line 262: these studies?

line 269: have been sufficient?

line 273: what are indications of measures??

line 287: sentence is not correct. Is the word how missing?
